# Greenland Iceberg Melt Variability from High-Resolution Satellite Observations

Ellyn M. Enderlin[1,2], Caroline J. Carrigan[2], William H. Kochtitzky[1,2], Alexandra Cuadros[3], Twila Moon[4], Gordon S. Hamilton[1,2,*]

[1]Climate Change Institute, University of Maine, Orono, ME, USA 04469
[2]School of Earth and Climate Sciences, University of Maine, Orono, ME, USA 04469
[3]School of Marine Sciences, University of Maine, Orono, ME, USA 04469
[4]National Snow and Ice Data Center, University of Colorado, Boulder, CO, USA 80303
*Deceased

*Correspondence to*: Ellyn M. Enderlin (ellyn.enderlin@gmail.com)

**Abstract.** Iceberg discharge from the Greenland Ice Sheet accounts for up to half of the freshwater flux to surrounding fjords and ocean basins, yet the spatial distribution of iceberg meltwater fluxes is poorly understood. One of the primary limitations for mapping iceberg meltwater fluxes, and changes over time, is the dearth of iceberg submarine melt rate estimates. Here we use a remote sensing approach to estimate submarine melt rates during 2011-2016 for 637 icebergs discharged from seven marine-terminating glaciers fringing the Greenland Ice Sheet. We find that spatial variations in iceberg melt rates generally follow expected patterns based on hydrographic observations, including a decrease in melt rate with latitude and an increase in melt rate with iceberg draft. However, we find no longitudinal variations in melt rates within individual fjords. We do not resolve coherent seasonal to interannual patterns in melt rates across all study sites, though we attribute a four-fold melt rate increase from March to April 2011 near Jakobshavn Isbræ to fjord circulation changes induced by the seasonal onset of iceberg calving. Overall, our results suggest that remotely-sensed iceberg melt rates can be used to characterize spatial and temporal variations in oceanic forcing near often inaccessible marine-terminating glaciers.

## 1 Introduction

The Greenland Ice Sheet discharges ~550 Gt of icebergs per year (Enderlin et al., 2014). This accounts for approximately a third to a half of the total freshwater flux from Greenland to the surrounding fjords and ocean basins (Bamber et al., 2012; Enderlin et al., 2014; van den Broeke et al., 2016). Unlike surface meltwater runoff fluxes from the ice sheet and tundra, which primarily enter the ocean system from point sources (subglacial discharge channels and terrestrial rivers, respectively), icebergs act as distributed freshwater sources. The spatial distribution of iceberg freshwater fluxes is dependent on a number of factors, including the volume and size distribution of ice calved from each glacier, which varies substantially over a range of spatial scales (Enderlin et al., 2014), and the solid-to-liquid conversion rate of an iceberg's freshwater reserves. Although surface sublimation and melting, wave erosion, and submarine melting all contribute to iceberg ablation, the solid-to-liquid conversion rate should primarily be dictated by submarine melting because of the strong dependence of total ablation on the surface area over which each process acts (e.g., Enderlin et al. (2016), Moon et al. (2017)).

Depending on the rate of submarine melting, the submerged surface area over which submarine melting occurs, and the residence time of icebergs in Greenland fjords, up to half of iceberg discharge can be converted to liquid freshwater before entering the open ocean (Mugford and Dowdeswell, 2010; Enderlin et al., 2016). The location where iceberg meltwater enters the ocean system is proving important for local to global ocean circulation (Luo et al., 2016; Stern et al., 2016), yet the spatial distribution of iceberg meltwater fluxes has been largely overlooked because it cannot be estimated from existing hydrographic observations (Jackson et al., 2016). Where iceberg residence times can be estimated from, for example, iceberg tracking (Sulak et al., 2017), these data can be paired with remotely-sensed iceberg size and area distributions (Enderlin et al., 2016; Sulak et al., 2017) and empirical iceberg melt rates to estimate iceberg freshwater fluxes. However, there are only a handful of locations around Greenland where there are sufficient water temperature and velocity records to constrain empirical iceberg melt rate estimates in iceberg-congested fjords (e.g., Bendtsen et al. (2015), Gladish et al. (2015), Jackson et al. (2016)). To address the dearth of iceberg melt rate estimates in Greenland's fjords, here we use a satellite remote sensing method to construct time series of submarine melt rates and meltwater fluxes for icebergs calved from seven large outlet glaciers spanning the periphery of the Greenland Ice Sheet (Fig. 1). Although the iceberg melt estimates constructed using this remote sensing method are limited to irregular observation periods during 2011-2016, the data provide the most comprehensive observationally-constrained estimates of Greenland iceberg melting to date.

## 2 Methods

As a freely-floating iceberg ablates, the elevation of its surface lowers in proportion to the iceberg's volume loss so that the iceberg remains in hydrostatic balance with the water in which it is submerged. This principle enables the estimation of iceberg meltwater fluxes (i.e., volume lost due to submarine melting per unit time) from repeat remotely-sensed surface elevation observations. Here we follow the approach of Enderlin and Hamilton (2014) to estimate changes in surface elevation using very high-resolution stereo satellite images acquired by the WorldView constellation of satellites. We note that this method could also be applied to elevation time series from terrestrial laser scanners, stereo imagery acquired by unmanned aerial vehicles or other satellite platforms, or GPS-derived elevations, but we focus on WorldView data because, unlike data acquired from the other platforms, WorldView data can be used to construct multi-year records of iceberg elevation change around the entire ice sheet periphery. Using this approach, we produce iceberg melt estimates from multiple observation periods during 2011-2016 (Fig. 1, Table 1) for seven large marine-terminating glaciers across southeast, northeast, and western Greenland that have sufficient WorldView image archives to estimate iceberg melt rates for more than one observation period.

For each study site, we used a combination of the Surface Extraction with TIN-based Search-space Minimization (SETSM) (Noh and Howat, 2015) and NASA Ames Stereo Pipeline (ASP) (Shean et al., 2016) to construct very high-resolution (2 m

horizontal resolution, ~3 m vertical uncertainty (Enderlin and Hamilton, 2014)) digital elevation models (DEMs) of iceberg-congested waters. A comparison of the DEMs produced using the SETSM and ASP algorithms indicates that the accuracy of iceberg elevations derived from the algorithms are comparable, allowing us to switch from the use of SETSM DEMs for 2011-2014 images to ASP DEMs for 2015-2016 images without biasing our results. DEMs were constructed over the entire

stereo image domain so that bedrock and water surface elevations could be used to co-register DEMs (Enderlin and Hamilton, 2014).

To estimate the change in iceberg volume between image acquisition dates, we applied the same DEM-differencing approach as Enderlin and Hamilton (2014) and Enderlin et al. (2016): changes in iceberg surface elevation were manually

extracted from repeat co-registered DEMs, then converted to estimates of iceberg volume change under the assumption of hydrostatic equilibrium. The contribution of iceberg surface melting to the observed volume change was estimated from the daily runoff time series for the nearest glaciated pixel in the Regional Atmospheric Climate Model (RACMO) for Greenland (van Meijgaard et al., 2008; van Angelen et al., 2014), then subtracted from the ice volume change estimates to yield ice volume loss due to submarine melting. Although there are slight differences in runoff estimates generated by RACMO v2.3

(used for 2011-2014) and v2.4 (used for 2015-2016), the version of RACMO used in our analysis had no appreciable influence on ice volume loss partitioning because volume loss due to surface melting constituted <5% of total volume change. We converted our estimates of ice volume lost via submarine melting to estimates of liquid freshwater flux (cubic meters of meltwater produced per day) and average submarine melt rates (meters per day) over the submerged iceberg areas. To estimate the average draft (i.e., keel depth) and submerged area of each iceberg, we assumed that the submerged iceberg

shapes can be approximated by cylinders with dimensions defined by the iceberg surface elevation and surface area estimates (Enderlin and Hamilton, 2014). Under this assumption, the draft ($d$) is estimated as

$$d = \frac{\rho_i}{\rho_{sw} - \rho_i} z, \tag{1}$$

and the area-averaged melt rate ($\dot{m}$) is estimated as

$$\dot{m} = \frac{\Delta V / \Delta t}{2\pi r d + \pi r^2}, \tag{2}$$

where $z$ is the median ice surface elevation, $\rho_i$ and $\rho_{sw}$ are the ice and sea water densities, respectively, $\Delta V$ is the change in volume between image acquisition dates, $\Delta t$ is the time between image acquisition dates, and $r$ is the average radius of the iceberg surface in each image pair. Submerged iceberg shapes are likely to be more complex than the cylindrical shapes used herein but are impossible to discern from surface observations alone. However, good agreement among iceberg melt rates derived via DEM-differencing and empirical melt rates estimates in Helheim's fjord (Enderlin and Hamilton, 2014;

FitzMaurice et al., 2016; Moon et al., 2017) suggests that submerged iceberg shapes can be reasonably approximated by cylinders.

Uncertainty in the submarine meltwater flux, submerged area, draft, and melt rate estimates are described in detail in Enderlin and Hamilton (2014) and are, therefore, only summarized briefly here. All errors are propagated through our calculations, then summed in quadrature. Potential errors arise from (1) surface elevation errors, (2) uncertainty in the operator-defined iceberg tracking, (4) uncertainties/changes in the ice and ocean water densities used to convert elevation change to volume change, (5) surface melt over- or under-estimation, and (6) changes in the iceberg surface area between image acquisitions. Systematic and random errors in iceberg elevations are minimized through vertical coregistration of iceberg DEMs using neighboring open water elevations and through spatial averaging, respectively. Uncertainties introduced by manual translation and rotation of iceberg masks in repeat DEMs are quantified through repeated delineation of each iceberg. Ice and water densities are assumed to vary by up to 10 kg m$^{-3}$ and 2 kg m$^{-3}$, respectively, between observations. A conservative surface meltwater uncertainty of 30% is applied to account for RACMO uncertainties and potential deviations in the melt rate of icebergs from the nearest glacierized RACMO grid cell. The surface area uncertainty is defined as the temporal range about the mean. The typical (i.e., median) uncertainties in the submarine meltwater flux, draft, submerged area, and melt rate are 25.6%, 2.7%, 3.2%, and 27.6%, respectively.

Deviations in iceberg shape from the assumed cylindrical geometry is not explicitly accounted for in our draft, submerged area, and melt rate uncertainty estimates. Our melt rate estimates assume that the iceberg shape changes uniformly over time even though empirical melt rate estimates suggest that melt rates vary with depth (e.g., Moon et al. (2017)), leading to unstable geometries and mechanical failure over longer time periods (e.g., Wagner et al. (2014)). We are, however, only considering iceberg geometry evolution over roughly monthly time scales. Empirical meltwater flux estimates (Moon et al., 2017) suggest changes in iceberg geometry are negligible over such short time periods. To demonstrate, we turn to previous estimates of iceberg melt rates for icebergs calved from Helheim Glacier and Jakobshavn Isbræ, where melt rates for large deep-drafted icebergs can vary by up to ~0.5 m/d from the surface to the iceberg base (Enderlin et al., 2016; Moon et al., 2017). Assuming a (simplified) linear increase in the melt rate from the surface to the iceberg base, the submerged area of a 500 m-wide and 350 m-deep iceberg would change by ~0.33% per day as its shape evolved from a cylinder to a cone. Thus, although we cannot quantify the potential systematic under-estimation of the submerged iceberg areas (and associated over-estimation of submarine melt rates) that results from the use of idealized submerged geometries, we are confident that any changes in iceberg submerged geometries over the timescales considered here are reasonably captured by our submerged area uncertainty estimates.

## 3 Results and Discussion

We extracted a total of 637 iceberg meltwater flux and melt rate estimates near the termini of seven large marine-terminating outlet glaciers fringing the Greenland Ice Sheet periphery and spanning March-October of 2011-2016 (Table 1; Enderlin, 2017). The number of estimates varies widely, with 3 to 27 melt estimates per observation period (mean=15). In general, the

number of estimates is inversely proportional to the distance between the icebergs and their parent glaciers and the time period between image acquisitions, restricting our analysis to icebergs located within ~10 km of the glacier termini and to time spans of 3-67 days.

### 3.1 Regional Patterns

In line with previous analyses of meltwater fluxes for icebergs calved from Helheim Glacier in the southeast (Enderlin and Hamilton, 2014) and Jakobshavn Isbræ in the west (Enderlin et al., 2016), we find that the meltwater flux generally increases with the submerged iceberg area (Fig. 2). Linear polynomials fit to all meltwater flux and submerged area estimates at each study site provide a means to quantify regional variations in the efficiency of iceberg melting around Greenland. Variations in the slope of the linear polynomial fit reflect regional differences in the rate of submarine melting (Fig. 2). The site-specific

meltwater flux area-based parameterizations, correlation coefficients, and root mean square error estimates are listed in Table 1. We generally find the highest melt rates near Koge Bugt and Helheim glaciers in the southeast (>0.35 m/d), with slightly lower melt rates in the Disko Bay (Jakobshavn) and Upernavik regions in the central west (~0.25-0.35 m/d). Icebergs calved from Alison and Kong Oscar glaciers in the Baffin Bay region in the northwest melt at slightly slower rates than those in the central west (~0.14-0.24 m/d). The lowest melt rates are found for icebergs calved from Zachariæ Isstrøm in the northeast

(~0.12 m/d).

The observed large-scale spatial patterns in melt rate generally follow expected variations based on regional differences in subsurface ocean temperatures (e.g., Straneo et al. (2012)) and surface meltwater runoff (e.g., van den Broeke et al. (2016)), which drives summertime fjord circulation (Jackson and Straneo, 2016). There are, however, some notable exceptions. The

average melt rate estimate for Koge Bugt is nearly double the average melt rate for icebergs calved from Helheim Glacier despite similar water temperatures near the fjord mouths (Sutherland et al., 2013). Although our Koge Bugt dataset includes only seven icebergs across two observation periods, we observe melt rates of >0.6 m/d during both observation periods, increasing our confidence that the difference in average melt rates reflects variations in typical melt conditions at the two study sites and is not due to observational uncertainties or anomalous melt conditions. We also find a discrepancy in the

predicted latitudinal decrease in the iceberg melt rates in northwest Greenland, where we observe lower melt rates for icebergs calved from Alison Glacier than the more northerly Kong Oscar Glacier. We hypothesize that the strengthened latitudinal gradient in the southeast and reversed gradient in the northwest are due to spatial variations in turbulent melting below the waterline associated with differences in near-surface water temperatures and/or relative velocity (i.e., difference in water and iceberg velocities) for icebergs located in kilometers-long iceberg-congested fjords (Helheim and Alison) versus

freely-floating icebergs in close proximity to the open ocean (Koge Bugt and Kong Oscar). Additional in situ water temperature and velocity observations are required to test this hypothesis, but if proven true, it suggests that near-terminus hydrography is strongly influenced by fjord geometry.

## 3.2 Local Patterns

Although detailed *in situ* hydrographic analyses of Greenland's glacial fjords are limited in space and time, existing observations indicate that there are much steeper gradients in water temperature and velocity in the vertical plane (i.e., with depth) than in the horizontal plane (i.e., along fjord) (Sutherland et al., 2014; Bendtsen et al., 2015; Gladish et al., 2015; Jackson et al., 2016). As such, we expect to find pronounced variations in melt rates for icebergs that do and do not penetrate into the relatively warm and salty water masses found below ~100-200 m-depth around the ice sheet periphery (Straneo et al., 2012; Moon et al., 2017) but no discernible variations in melt rates with distance from the parent glacier.

To examine the depth-dependency of iceberg melt rates, we first sorted the icebergs according to their median draft. After parsing the icebergs into 50 m-increment draft bins, we calculated the medians of all the area-averaged melt rate estimates (hereafter the median melt rate) and draft estimates in each bin. Figure 3 shows the binned median melt rates and drafts for each study site. For all study sites, the median melt rates are generally smaller for icebergs in the upper ~200 m of the water column than those that penetrate to greater depths (Fig. 3a, Figs. 3b-h). The depth-dependency of iceberg melt rates is particularly pronounced for icebergs calved from the Upernavik glaciers (Fig. 3d) and Jakobshavn Isbræ (Fig. 3e) in the central west. For Upernavik, the median melt rate increases from the surface down to ~150 m-depth, decreases slightly over the 150-200 m depth range, then increases again below 200 m-depth. For Jakobshavn, the median melt rate increases from the surface down to ~150 m-depth, decreases down to ~250 m-depth, then increases again down to 350 m-depth. The apparent decrease in the melt rate below 350 m-depth reflects one observation from March 2011 when melt rates were particularly low, as discussed more below. Although the dip in melt rates at ~200 m-depth is not significant (i.e., does not exceed the uncertainty of neighboring bins), it coincides with the approximate depth of the interface between the colder near-surface waters and warmer sub-surface waters observed in Jakobshavn's fjord (Ilulissat Isfjord) (Gladish et al., 2015) and the Upernavik fjord system (Fenty et al., 2016), where water velocities should be relatively slow and turbulent melting should reach a local minimum (Moon et al., 2017). These observations suggest that our remote sensing method may be capable of resolving the depth of the near- and sub-surface water interface where hydrographic observations are difficult or impossible to acquire, such as near the termini of calving glaciers. However, we caution that the area-averaged melt rates obtained using this approach likely under-estimate the trend of increasing melt rates with depth because of the integrative nature of our area-averaged melt rate estimates.

## 3.3 Temporal Patterns

The stratification and circulation of water masses near Greenland's glacier termini likely vary over weekly to inter-annual time scales with changes in wind direction (Jackson et al., 2014), glacial meltwater discharged from the base of the glacier termini (Mortenson et al., 2011; Cowton et al., 2015), sea ice/ice mélange extent (e.g., Enderlin et al. (2016) and Shroyer et al. (2017)), and the properties of water masses advected along the continental shelf (Holland et al., 2008; Mortenson et al.,

2011). To investigate potential temporal variations in iceberg melt rates, we parsed our observations according to their observation periods and computed the median melt rate and median draft for each draft bin over the individual observation periods (Fig. 4). Our data suggest that across all study sites there were neither substantial seasonal nor inter-annual changes in melt rate during 2011-2016, though limited observations from Jakobshavn's fjord (discussed below) demonstrate that the lack of a coherent temporal signal across all study sites does not preclude the existence of temporal variations.

Our finding that, overall, there is no seasonal or inter-annual variation is in contrast to empirical melt estimates, which suggest there should be pronounced seasonal differences in iceberg melt rates (Mugford and Dowdeswell, 2010) primarily due to the strong dependency of iceberg melting on water velocities (Bigg et al., 1997; FitzMaurice et al., 2016; FitzMaurice et al., 2017). The lack of substantial coherent temporal variability in our iceberg melt rate estimates may be influenced by a number of factors. First, the number of repeat DEMs and timing of DEM acquisitions varies substantially from year-to-year and between study sites, making it difficult to infer seasonal and inter-annual patterns from our dataset. Second, our remotely-sensed melt rates integrate variations in melt rate with depth and over the time interval between DEM acquisition dates. The depth integration likely has little influence on shallow-drafted icebergs that are bathed in relatively homogeneous water but may substantially reduce the melt rates for deep-drafted icebergs, as previously mentioned. The time-integrative nature of our remotely-sensed melt rates means that high-frequency variations in iceberg melting are smoothed-out. Temporal smoothing is likely to be particularly important during the seasonal transition from winter conditions (i.e., expansive sea ice, little subglacial meltwater discharge, synoptic-scale changes in fjord circulation) to summer conditions (i.e., open water with fjord circulation driven by subglacial discharge) (Jackson et al., 2014), which may lead to rapid changes in submarine melt rates. Finally, uncertainties in the melt rate estimates introduced by observational uncertainties, particularly uncertainty in the submerged iceberg shape, may also partially obscure temporal variations in iceberg melting over seasonal to inter-annual time scales. While our results here validate our use of time-averaged melt rates in the spatial analyses presented above, further research on temporal variations in iceberg melt is necessary to determine whether changes in iceberg meltwater fluxes over time have an appreciable impact on local-to-regional ocean circulation, motivating the need for more detailed time series of iceberg melt rates around Greenland.

Despite the limited ability of our remotely-sensed iceberg melt estimation method to detect seasonal to inter-annual iceberg melt rate variations over the relatively long, irregular observation periods typically available from WorldView DEMs, our results indicate that the method is capable of detecting abrupt changes in iceberg melting when the DEM repeat interval is short and coincides with large changes in iceberg melt conditions. Melt rates compiled for icebergs calved from Jakobshavn Isbræ indicate that there was a nearly four-fold increase in deep-drafted iceberg melt rates in Ilulissat Isfjord between late March and early April 2011 (Fig. 5). This rapid increase in iceberg melting coincided with the appearance of distinct lateral shear margins in the April 6th WorldView image of the fjord's extensive ice mélange, which were not present in a March 19th WorldView image. Surface air temperatures observed at the closest on-ice automatic weather station (673 m a.s.l.;

67.097°N, -49.933°E) lapsed to sea level indicate that regional air temperatures were well below freezing (daily mean temperatures <-10°C) for 20 of 24 days between the image acquisitions; thus, the appearance of the shear margins cannot be easily explained by surface melting. We suggest that shear margins instead appeared as a result of abrupt mélange motion away from the terminus during a large calving event. Seismic data recorded in Ilulissat, at the fjord mouth, confirm that the

earliest large-scale calving event of 2011 occurred on April 3rd, 3 days prior to the beginning of our second observation period.

Based on the large change in deep-drafted melt rates and coincident onset of seasonal calving, we hypothesize that iceberg over-turning during the calving event altered the stratification and circulation of the fjord water masses, which rapidly

increased iceberg melt rates at depth. Although the size of the calving event and the degree of mixing within the water column are unknown, laboratory experiments of iceberg over-turning indicate that the amount of energy released during a large calving event is far more than enough to entirely mix the water column within 1 km of Jakobshavn's terminus (Burton et al., 2012). To assess whether mixing-induced changes in water temperature or velocity was the more likely driver of the observed change in melt rate, we turn to the thermodynamic equations of submarine melting. Turbulent melting due to

horizontal water shear past an iceberg is estimated as

$$\dot{m}_{turbulent} = 0.58 v^{0.8} \frac{T_{sw} - T_i}{L^{0.2}},$$     (3)

and buoyancy-driven melting is

$$\dot{m}_{buoyant} = (7.62{\times}10^{-3})T_{sw} + (1.29{\times}10^{-3})T_{sw}^2,$$     (4)

where $v$ is the relative water velocity (i.e. water velocity with respect to the iceberg velocity), $T_{sw}$ and $T_i$ are the temperature

of the sea water and ice, respectively, and $L$ is the iceberg length. In the absence of changes in relative velocity, variations in water temperature within the range observed in Ilulissat Isfjord (Gladish et al., 2015) are insufficient to drive the four-fold increase in deep-drafted melt rates. However, for an ice temperature of -5°C (Vieli and Nick, 2011) and a water temperature of 2°C (Gladish et al., 2015), the relative velocity would need to increase from an average of approximately 0.06 m/s to 0.31 m/s to increase the turbulence-driven melt rate of large (~500 m-long) icebergs from ~0.12m/d to ~0.46m/d. The persistent

ice mélange near the Jakobshavn terminus prevents acquisition of the water temperature and velocity time series required to test this hypothesis. However, water velocity data from Sermilik Fjord in southeast Greenland suggest that velocities of ≥0.3 m/s (Jackson et al., 2014) are possible in Greenland's deep glacial fjords. Moreover, given the mostly below-freezing air temperatures observed over this period of rapid change, it is unlikely that the inferred changes in fjord circulation were triggered by the seasonal onset of glacier meltwater-enhanced subglacial discharge at depth in the fjord. Therefore, we

interpret the four-fold increase in melt rates as an indication that full-thickness calving events from large glacier termini may significantly alter the hydrographic properties of Greenland's glacial fjords, with a measurable influence on iceberg melt.

**4 Conclusions**

Here we apply a remote sensing method to construct submarine melt rate and meltwater flux time series for icebergs calved from seven large marine-terminating outlet glaciers spanning the Greenland Ice Sheet edge. We find that for each study site, the meltwater flux from icebergs can be reasonably approximated as a linear function of the submerged iceberg area. Differences in the rate of iceberg melting between study sites generally follow expected geographic patterns based on variations in ocean temperature and surface meltwater runoff from the ice sheet, with the highest melt rates in the southeast, decreasing melt rates with increasing latitude along the west coast, and the lowest melt rates in the northeast. We hypothesize that deviations from the expected latitudinal patterns are due to variations in the prevalence of icebergs and/or near-terminus water circulation associated with different fjord geometries, emphasizing the potential importance of Greenland fjord geometry on iceberg (and glacier) melt rates.

At finer spatial scales, our observations support the expected depth-dependency of iceberg melt rates in the highly-stratified water fringing Greenland: at each study site, melt rates are low and fairly uniform down to ~200 m-depth then gradually increase down to ~350 m below the sea surface. Although our melt rate time series across all study sites do not reveal coherent temporal variations in melting, observations compiled for Jakobshavn Isbræ's fjord suggest that abrupt changes in melt conditions do occur. Furthermore, these changes at depth can potentially be monitored using the remote sensing approach applied here. The data compiled for Jakobshavn Isbræ also suggest that full-thickness calving events may be important for fjord circulation and iceberg melt, though additional melt rate estimates with ~weekly temporal resolution, possibly from terrestrial laser scanner or unmanned aerial vehicle observations, are required to test the effect of calving on sub-surface melt conditions.

Overall, we conclude that the DEM-differencing approach provides an excellent means to quantify spatial variations in iceberg melting and potentially resolve rapid temporal changes in iceberg melting when elevation observations with short repeat intervals are available. Quantification of iceberg melt rates around Greenland, and beyond, will enable the construction of more accurate ice sheet freshwater flux boundary conditions in ocean models and an improved understanding of the impacts of terrestrial ice mass loss on ocean circulation. Furthermore, if spatial and temporal patterns in iceberg melting can be linked to variations in water temperature and/or velocity, then remotely-sensed iceberg melt rates may be useful for inferring changes in iceberg and glacier melt conditions in glacial fjords in the absence of in situ hydrographic observations.

**Data Availability**

The location, median surface elevation, surface elevation uncertainty, and vertical co-registration for each observation date and estimates of the ice volume change rate, uncertainty in the ice volume change rate, average draft, range in draft, average

surface area, range in surface area, average submerged area, and range in submerged area between observation dates for all icebergs in our analysis can be accessed at https://arcticdata.io/catalog/#view/doi:10.18739/A20N7C.

## Author Contributions

E.M.E. developed the methods used to extract iceberg melt data from WorldView digital elevation models, extracted data for two study sites, supervised data extraction performed by coauthors, compiled and analyzed the data, and wrote the manuscript. C.J.C., W.H.K., and A.C. compiled satellite images, constructed digital elevation models, and extracted iceberg melt data for five study sites. T.M. assisted with manuscript preparation and revisions. G.S.H. assisted with method development.

## Acknowledgements

This paper is dedicated to Gordon Hamilton, who helped develop the DEM-differencing approach used to construct the iceberg melt time series. This work was supported by National Science Foundation Arctic Natural Sciences grant 1417480 to E. M. Enderlin and National Science Foundation Graduate Research Fellowship Program grant DGE-1144205 to W. H. Kochtitzky. WorldView images were distributed by the Polar Geospatial Center at the University of Minnesota (http://www.pgc.umn.edu/imagery/satellite/) as part of an agreement between the US National Science Foundation and the US National Geospatial Intelligence Agency Commercial Imagery Program. RACMO Greenland v2.3 runoff data for 2011-2014 and v2.4 runoff data for 2015-2016 were provided by Dr. Michiel van den Broeke, Utrecht University (https://www.projects.science.uu.nl/iceclimate/models/greenland.php). The AOTIM-5 tidal model used for DEM co-registration was obtained from https://www.esr.org/research/polar-tide-models/list-of-polar-tide-models/aotim-5/. Automated weather station data for Jakobshavn Isbræ were obtained from the Programme for Monitoring of the Greenland Ice Sheet (PROMICE; http://promice.org/WeatherStations.html). Seismic data from Ilulissat were provided by Dr. Jason Amundson, University of Alaska Southeast.

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

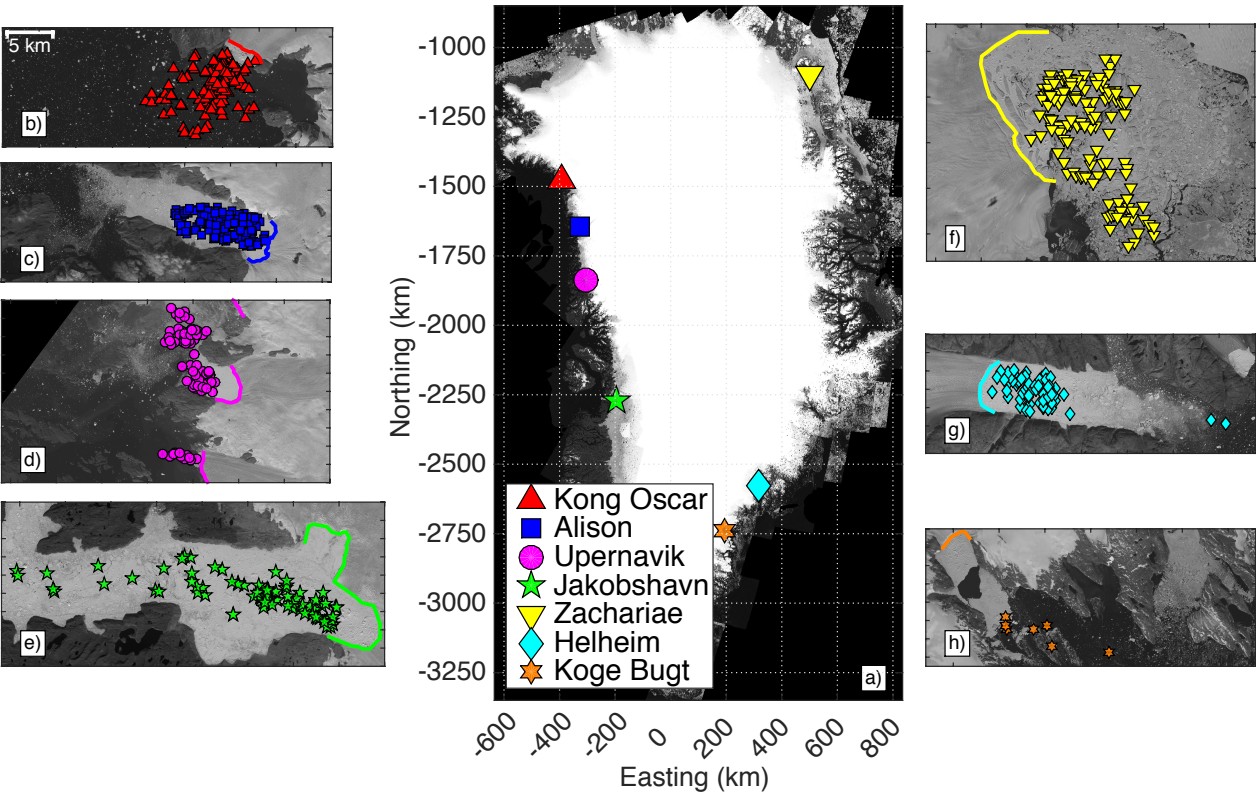

**Figure 1: Location of Greenland icebergs included in this study. (a) The locations of the glaciers from which the icebergs calved overlain on the GIMP image mosaic. The different iceberg sources are distinguished by symbol color and shape (see legend). (b-h) Locations of all study icebergs overlain on summer 2016 Landsat 8 panchromatic images of (b) Kong Oscar Glacier, (c) Alison Glacier, (d) Upernavik Glacier, (e) Jakobshavn Isbræ, (f) Zachariæ Isstrøm, (g) Helheim Glacier, and (h) Koge Bugt Glacier. The same scale, shown in panel b, is used in panels b-h. Termini of the icebergs' glacier sources are delineated with colored lines in panels b-h.**

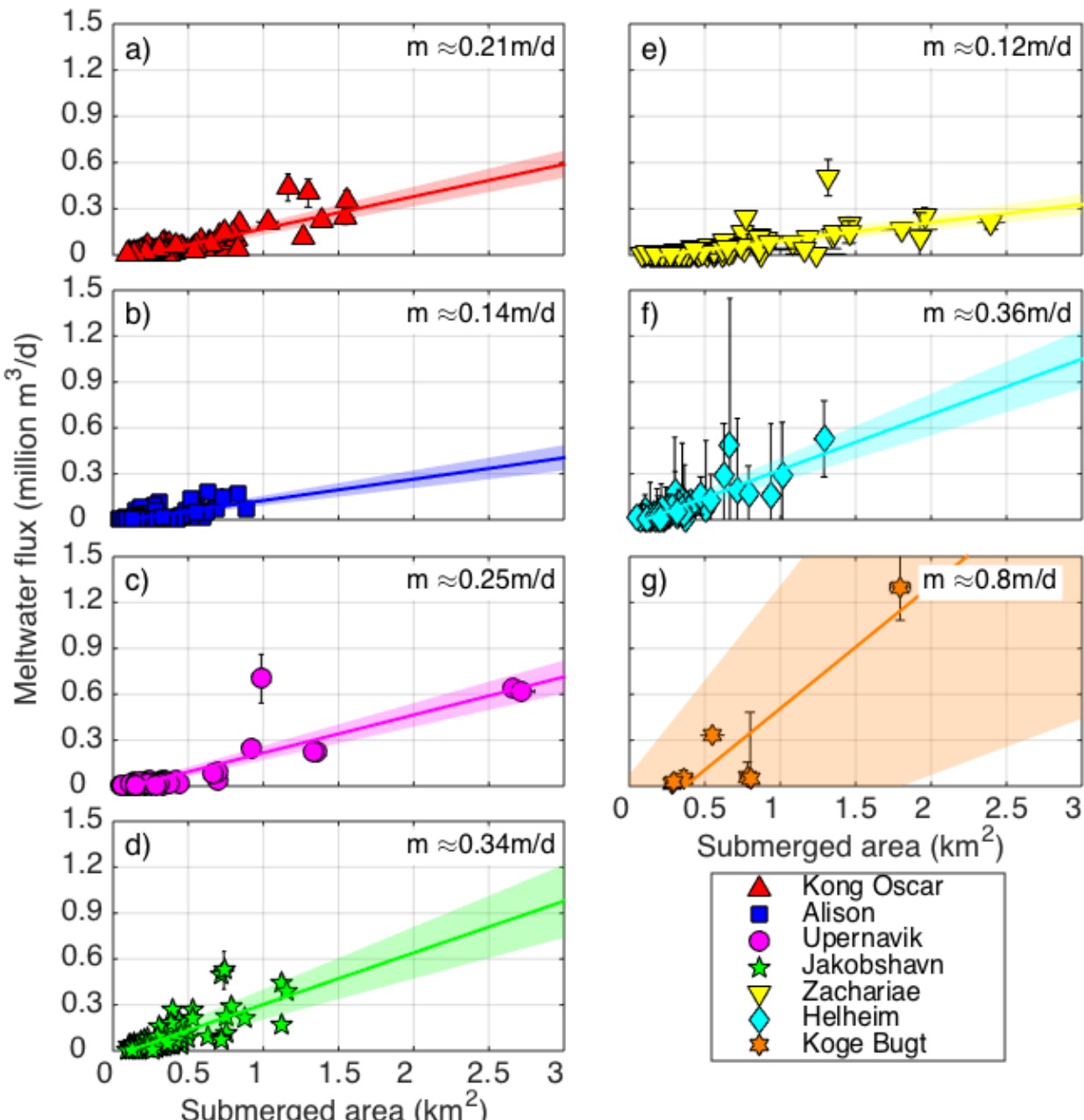

**Figure 2: Liquid freshwater fluxes (millions of cubic meters per day) plotted against the estimated submerged area (square kilometers) for all icebergs sampled near the terminus of (a) Kong Oscar Glacier, (b) Alison Glacier, (c) Upernavik Glacier, (d) Jakobshavn Isbræ, (e) Zachariæ Isstrøm, (f) Helheim Glacier, and (g) Koge Bugt Glacier. Vertical error bars indicate the meltwater flux uncertainties due to random DEM errors, ice density uncertainties, surface meltwater flux uncertainties, and manual iceberg delineation errors. Horizontal error bars indicate the range of submerged iceberg areas predicted for cylindrical submerged geometries using surface elevation and map-view surface area estimates extracted from repeat DEMs. Linear polynomials fit to the datasets compiled for each study site are plotted as thick colored lines and the surrounding shaded envelopes encompass their 95% confidence intervals. Area-averaged submarine melt rates derived from the polynomials are listed in each panel.**

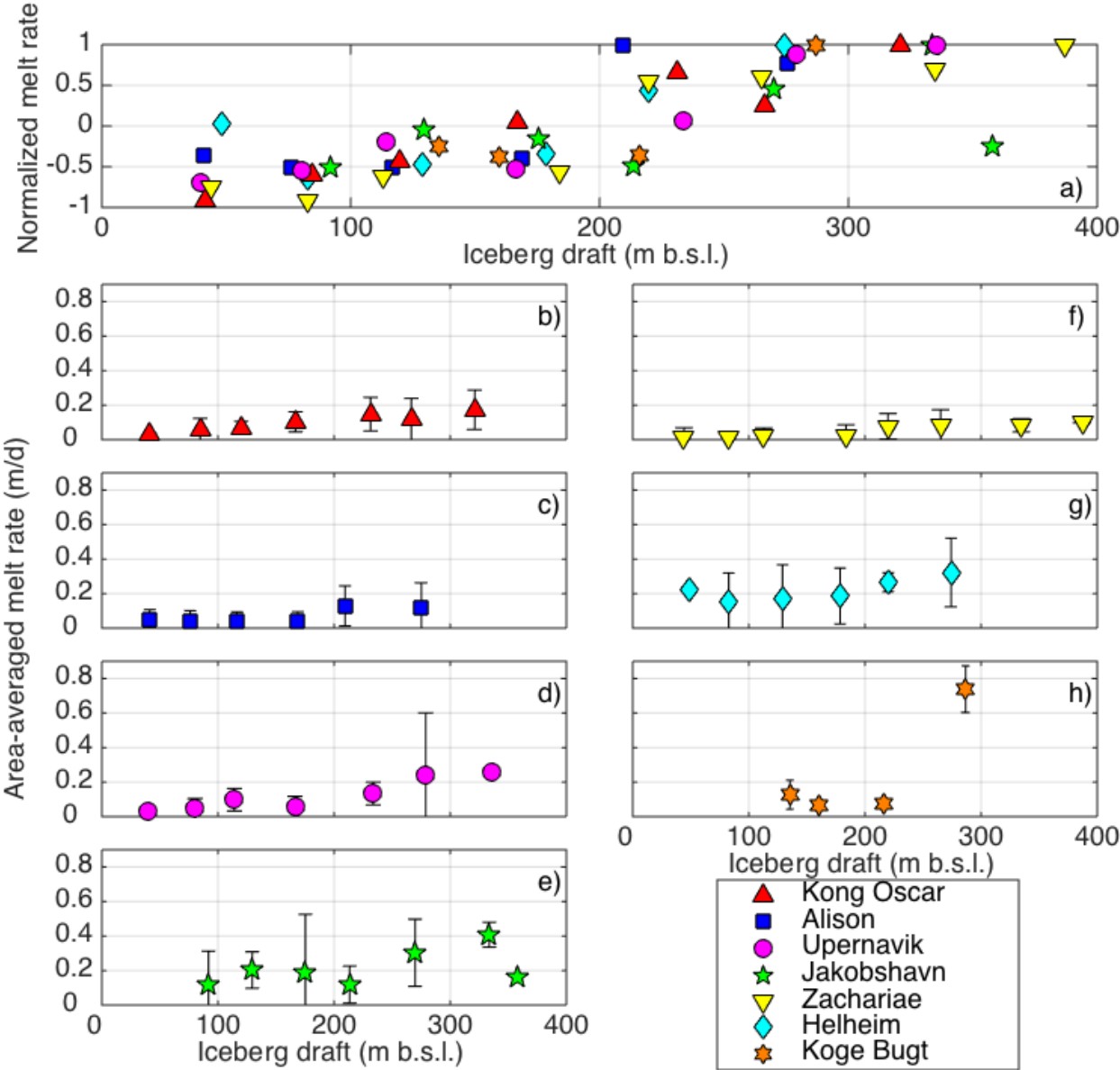

**Figure 3: Plots of melt rate variability with draft. (a) Normalized melt rate plotted against median draft (meters below sea level). Normalized melt rates less than zero are below the observed average and values greater than zero indicate above-average melt rates. (b-h) Area-averaged melt rate (meters per day) plotted against median draft (meters below sea level) for icebergs near the terminus of (b) Kong Oscar Glacier, (c) Alison Glacier, (d) Upernavik Glacier, (e) Jakobshavn Isbræ, (f) Zachariæ Isstrøm, (g) Helheim Glacier, and (h) Koge Bugt Glacier. In all panels, icebergs are sorted into 50 m-increment draft bins and the symbols mark the median values for each draft bin. In (b-h), vertical error bars bound the range of melt rates.**

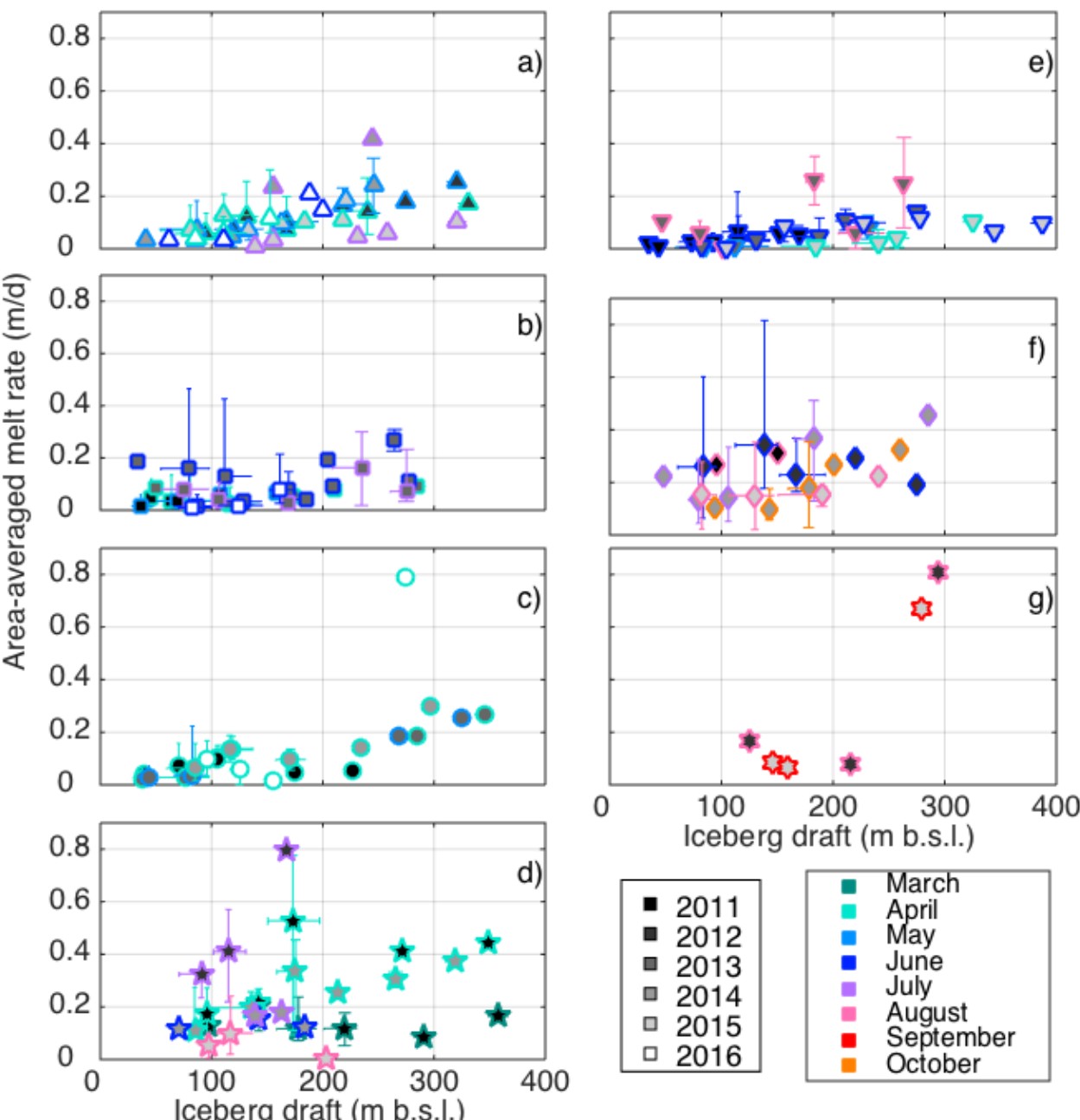

**Figure 4: Area-averaged iceberg melt rate (meters per day) plotted against median draft (meters below sea level) for icebergs sampled near the terminus of (a) Kong Oscar Glacier, (b) Alison Glacier, (c) Upernavik Glacier, (d) Jakobshavn Isbræ, (e) Zachariæ Isstrøm, (f) Helheim Glacier, and (g) Koge Bugt Glacier. For each observation period, icebergs were organized into 50 m-deep draft bins and the median melt rate and draft were computed. The symbols mark the median values and the error bars mark the range of estimates for each draft bin. The face colors and edge colors of the symbols indicate the year and month of the observations, respectively (see legends).**

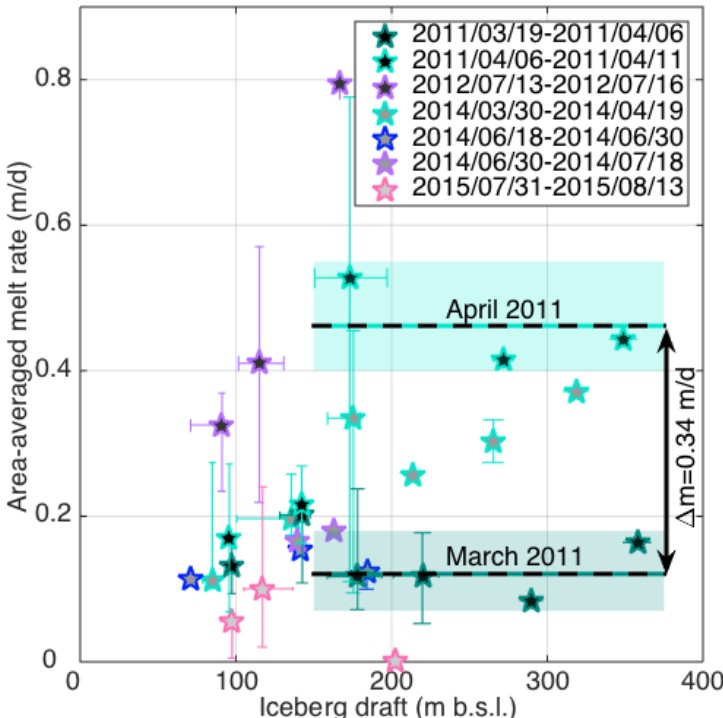

**Figure 5: Area-averaged submarine melt rates plotted against median draft for icebergs calved from Jakobshavn Isbræ, west Greenland, into Ilulissat Isfjord. As in Figure 4, the symbols mark the median values and the error bars mark the range of estimates for each draft bin. The face colors and edge colors of the symbols indicate the year and month of the observations, respectively, as described in the legend. The large increase in the area-averaged melt rate below 150 m-depth from March to April in 2011 are highlighted by the shaded rectangles. The dashed lines within the rectangles mark the average melt rates for icebergs with drafts >150 m and the difference in the average melt rate between observation periods is denoted by the double-sided arrow.**

| Glacier | Year | Period | Observations | Observations per 50 m-draft bin | ΔV/Δt = f(Asub) |
|---|---|---|---|---|---|
| Kong Oscar | 2012 | 04/14-04/26 | 22 | 0, 5, 6, 5, 4, 0, 1, 0 | 0.209A-38413 (R = 0.85 & RMSE = 40447) |
| | | 04/26-06/11 | 20 | 0, 0, 7, 6, 4, 1, 1, 0 | |
| | 2014 | 05/04-06/13 | 18 | 1, 7, 7, 0, 3, 0, 0, 0 | |
| | | 06/13-08/04 | 3 | 0, 0, 0, 1, 1, 0, 0, 0 | |
| | 2015 | 04/06-04/19 | 13 | 0, 6, 4, 1, 1, 0, 0, 0 | |
| | | 05/12-06/11 | 13 | 0, 1, 8, 2, 1, 0, 0, 0 | |
| | | 06/11-08/12 | 7 | 0, 0, 2, 1, 1, 1, 1, 0 | |
| | 2016 | 03/18-05/19 | 16 | 0, 4, 8, 3, 0, 0, 0, 0 | |
| | | 05/19-07/20 | 4 | 0, 1, 1, 1, 1, 0, 0, 0 | |
| Alison | 2011 | 03/25-04/11 | 19 | 5, 8, 5, 0, 0, 0, 0, 0 | 0.140A-13878 (R = 0.69 & RMSE = 24120) |
| | | 04/11-06/10 | 24 | 7, 13, 2, 1, 0, 0, 0, 0 | |
| | 2013 | 04/10-05/12 | 16 | 1, 6, 4, 2, 1, 1, 0, 0 | |
| | | 05/12-06/22 | 18 | 0, 5, 6, 3, 1, 2, 0, 0 | |
| | | 06/22-07/07 | 25 | 2, 10, 7, 2, 1, 2, 0, 0 | |
| | | 07/07-07/22 | 22 | 0, 6, 7, 2, 3, 3, 0, 0 | |
| | 2016 | 05/08-07/14 | 16 | 0, 4, 8, 4, 0, 0, 0, 0 | |
| Upernavik | 2011 | 03/26-04/12 | 21 | 3, 10, 5, 1, 1, 0, 0, 0 | 0.248A-31052 (R = 0.88 & RMSE = 61611) |
| | 2013 | 04/12-04/30 | 17 | 4, 6, 4, 0, 0, 1, 1, 0 | |
| | | 04/30-06/03 | 16 | 5, 8, 0, 0, 0, 1, 1, 0 | |
| | 2014 | 03/28-04/17 | 20 | 0, 10, 4, 2, 2, 1, 0, 0 | |
| | 2016 | 04/16-04/27 | 9 | 0, 2, 5, 1, 0, 1, 0, 0 | |
| Jakobshavn | 2011 | 03/19-04/06 | 22 | 0, 2, 3, 9, 5, 1, 0, 1 | 0.338A-34434 (R = 0.74 & RMSE = 74454) |
| | | 04/06-04/11 | 14 | 0, 2, 2, 7, 0, 1, 1, 0 | |
| | 2012 | 07/13-07/16 | 13 | 0, 4, 7, 1, 0, 0, 0, 0 | |
| | 2014 | 03/30-04/19 | 19 | 0, 3, 6, 5, 1, 2, 1, 0 | |
| | | 06/18-06/30 | 6 | 0, 1, 1, 3, 0, 0, 0, 0 | |
| | | 06/30-07/18 | 3 | 0, 0, 1, 1, 0, 0, 0, 0 | |
| | 2015 | 07/31-08/13 | 8 | 0, 3, 4, 0, 1, 0, 0, 0 | |
| Zachariae | 2011 | 05/31-06/08 | 21 | 1, 4, 14, 1, 0, 0, 0, 0 | 0.118A-23772 (R = 0.75 & RMSE = 43816) |
| | | 06/08-07/10 | 24 | 2, 6, 14, 1, 0, 0, 0, 0 | |
| | 2013 | 04/01-06/05 | 23 | 0, 8, 9, 3, 2, 0, 0, 0 | |
| | | 06/05-07/25 | 27 | 0, 12, 2, 5, 6, 1, 0, 0 | |
| | | 07/25-08/10 | 15 | 1, 2, 2, 2, 5, 2, 0, 0 | |
| | 2015 | 04/01-05/01 | 17 | 0, 0, 0, 9, 3, 3, 1, 0 | |
| | | 06/02-07/02 | 9 | 0, 0, 1, 3, 2, 1, 1, 1 | |
| Helheim | 2011 | 08/21-08/24 | 3 | 0, 1, 0, 1, 0, 0, 0, 0 | 0.363A-37746 (R = 0.84 & RMSE = 53704) |
| | 2012 | 06/24-06/29 | 18 | 0, 5, 6, 4, 1, 1, 0, 0 | |
| | 2014 | 07/02-07/31 | 20 | 1, 7, 7, 3, 0, 1, 0, 0 | |
| | | 10/16-10/30 | 14 | 0, 1, 4, 6, 1, 1, 0, 0 | |
| | 2015 | 08/10-08/16 | 16 | 0, 3, 8, 4, 1, 0, 0, 0 | |
| Koge Bugt | 2012 | 08/09-08/13 | 3 | 0, 0, 1, 0, 2, 1, 0, 0 | 0.803A-295723 (R = 0.91 & RMSE = 219609) |
| | 2015 | 08/30-09/16 | 3 | 0, 0, 1, 1, 0, 1, 0, 0 | |

**Table 1: Overview of iceberg observations and derived melt rate parameterizations for each study site. (column 1) Glacier names, (columns 2-3) observation periods, (column 4) number of observations, (column 5) number of observations per 50 m-draft bin for each observation period (listed from 0-50 m to 350-400 m depths), and (column 6) meltwater flux parameterizations. In column 5, the correlation coefficients and root mean square error estimates for the linear area-based meltwater flux parameterizations are also provided.**