# Peer review of "Greenland Iceberg Melt Variability from High-Resolution Satellite Observations"

_The Cryosphere, 2017_

## Referee Comment (RC1) · J.M. Amundson (Referee) · 22 Oct 2017

Summary:

In this study the authors use digital elevation models derived from satellite imagery to investigate temporal and spatial observations in iceberg melt rates. The paper builds on previous work to provide estimates of iceberg melt rates across several fjords in Greenland, and demonstrates that iceberg melt rates depend on iceberg draft – with the caveat that draft and melt rates are inferred from calculations of subaerial volume and assumed iceberg geometry. Meltwater from icebergs appears to be an important source of freshwater for fjords in Greenland, and therefore this study has important implications for fjord circulation and submarine melting of glacier termini.

[Figure]

Major comments:

Most of my concerns with this paper are related to understanding the numerous sources of uncertainty that are inherent (and unavoidable) in the authors' calculations. It wasn't until I went back to re-read Enderlin and Hamilton (2014) that I realized that these uncertainties had already been addressed in some detail previously. Therefore I think this paper would benefit from a 1-2 paragraph summary of the sources of error and their impact on the melt flux and melt rate calculations. Presumably this summary would be in Section 2. My sense is that the error in the melt flux calculations is small and that those calculations are therefore pretty robust. The depth-averaged melt rate calculations are more tenuous because they rely on an assumed iceberg geometry, which affects both the submerged surface area and the iceberg draft.

I would also feel more comfortable with the discussion of how submarine melt rates vary with depth if the paper more explicitly referred to the depth-averaged melt rates and drafts as proxies. For example, I believe that the draft is calculated with something like:

$h' = $ "draft proxy" $= V_{sa}/A * \rho_i/(\rho_w-\rho_i)$,

where $h'$ is the draft proxy, $V_{sa}$ is the subaerial volume, A is the cross-sectional area of the iceberg at the waterline, and $\rho_i$ and $\rho_w$ are the densities of ice and water. Including something along these lines would more precisely indicate what is actually being plotted in the various figures. Something similar could be written out to describe the proxy for the depth-averaged melt rate. Not sure if this correct, but:

$m' = 1 / (h' *2*pi*r) * dV_{sm}/dt$,

where $m'$ is the proxy for the depth-averaged melt, r is the average radius at the water line, and $V_{sm}= V_{sa}*\rho_i/(\rho_w-\rho_i)$ is the submerged volume. Depending on exactly how this calculation is made, you may be able to cancel out some terms.

My concern is that I'm just not sure how much faith to put in the melt rate vs. draft

figures. There seems to be a pretty nice relationship between meltwater flux and sub-merged area (based on the assumed geometry). Is there a similarly nice relationship between meltwater flux and submerged volume? If not, maybe that can somehow be used to justify the choice of iceberg shape.

Also, in the discussion of the observed changes in melt rates at Jakobshavn (page 7), it would be nice to have the calculations of melt rates spelled out in more detail. How did you calculate the change in water velocity that would be needed to increase the melt rate? By how much would you have to change the water temperature to get a similar change in melt rate? Can you exclude our potential sources, such as an incursion of warm water? Did the melange remain intact following the calving event?

Minor comments:

p. 1, line 16: Consider pointing out in the abstract that you don't observe longitudinal variations in melt rates. That seems to be a pretty important finding.

p. 1, line 30: Sublimation also contributes to ablation.

p. 2, line 12: Why these seven glaciers?

p.2, line 4: Consider citing Alon Stern's JGR paper: The effects of Antarctic iceberg calving-size distribution in a global climate model

p. 3, line 1: Why the switch in processing schemes?

p. 3, line 22: Are any of the icebergs tabular?

p. 5, line 22: Perhaps cite John Mortenson's paper: Heat sources for glacial melt in a sub-arctic fjord (godthabsfjord) in contact with the Greenland ice sheet

---

## Referee Comment (RC2) · Anonymous Referee #2 · 13 Nov 2017

This paper investigates iceberg submarine melt variability in fjords for icebergs calved from seven large tidewater glaciers around the Greenland coast between 2011-16. The paper uses a method developed and presented previously in a detailed paper (Enderlin and Hamilton 2014) and utilises Worldview Imagery to generate iceberg DEMs. The paper shows clearly how the estimated iceberg submarine melt-rates show distinct melt patterns that one would expect based both on hydrographic observations and variations in latitude and iceberg draft. In the main, the paper is very clearly written and the findings are well supported by the analyses and data while the conclusion provides a very succinct and clear summary of the paper highlighting the key findings and the considerable potential of the method utilised.

There are a few areas where the paper is a little unclear and these are outlined below;

[Figure]

in particular, some changes to the Figures would improve the clarity of the paper. On occasions, just a little more text is needed to aid the reader and any such additions will not detract from the paper as it is not overly long.

P1, l17 – I was a somewhat unclear what you meant in the abstract when stating that you do not resolve "coherent" temporal variations in melt rates. After reading the paper, this became clearer but I think it makes sense to state more clearly here in the abstract that you do resolve coherent 'seasonal or interannual patterns in your iceberg melt-rates'.

P1, l27 – worth adding that the size distribution of the calved icebergs as well as the volume calved is crucial to the spatial distribution of iceberg freshwater fluxes.

P2, l17-18 – important to add the caveat here that this is true as long as the iceberg in question is floating.

P2, l21-23 – I think that you should also add other satellite platforms to your list of additional potential methods that could be used to derive elevation time-series.

P2, l31-32 – You state that "A comparison of the DEMs produced using the SETSM and ASP algorithms indicates that the accuracy of iceberg elevations is unaffected by the choice of the algorithm used to construct DEMs". You have presumably carried out some kind of analysis to demonstrate that this is the case; in which case, it would be beneficial to report briefly (even in one line) what "unaffected" means by referring to one example of the results derived from analysis on one of your iceberg data-sets.

Fig. 1. Each location map (insets b – h) needs a scale (unless the scale is the same in all of them in which case a scale is still needed somewhere). Furthermore, it would help to show clearly where the calving fronts of the glaciers are; this may be obvious in some figures either from visual clarity (g) or site familiarity (e) but for many, especially d, f and g, it's not really clear where the icebergs have come from. (I might add that it does become clear when I enhance the scale on the pdf to 400%, as the images are

very high quality, they are just very small at the resolution of the current figure). And this information is needed to help make sense of the text on P3, l26-29.

Furthermore, in Figure 1 and in all subsequent figures, I think you should re-order the glacier legend box and symbols from b) to h) i.e. Kong Oscar, Alison, Upernavik etc ending with Koge Bugt rather than alphabetically as currently which is much more confusing for the reader.

Fig. 2 and p4, l6. It would be very helpful to include the estimated submarine melt rates, derived from each linear polynomial, within the individual figure boxes (a – g).

P4, L16. Ref. should be to Jackson and Straneo, not et al.

P4, l18 – there are seven plotted symbols in Figure 1h) associated with Koge Bugt, not the six suggested in the text. Why the discrepancy?

P4, l21. The confidence for the Koge Bugt datasets, as currently explained, seems a little misplaced given the sample size. In particular, from Fig 1, it looks as though one of the icebergs sampled is a considerable distance from the others and perhaps in more open waters 'atypical' of the other fjord samples. As such, are you sure you are observing "typical melt conditions" and not getting spurious results due to anomalous sampling (particularly given the small sample size and thus significance of one anomalous data point to your overall results for KB).

P4, l31 – will detailed in-situ data become more widely available as part of the OMG programme?

P5, l8 – The results for individual glaciers (b-h) would be of much more use if the y scale was reduced from 0 – 0.4 m/d as opposed to 1m (with the exception of Koge Bugt) so that the (valuable) details in the variable melt-rates could be seen more clearly.

P5, L12-13. The broad description relating to melt rate with iceberg draft is not really correct when integrating the Upernavik and Jakobshavn Isbrae results. The melt rate actually continues to decrease in draft bin 200-250m at JI, not "increase" again as sug-

Interactive
comment

gested in L13. Hence the dip is broader than the 150-200m dip that you suggest. I think you just need to be a bit broader in your depth categorisation for the "approximate depth of the interface" for the cold-warm boundary (and I presume that it does vary between fjord systems). Furthermore, the melt rate appears to dip again at JI in the 350-400 m bin (actually dropping back to shallow 'cold' water values). Given the integrative nature of your area averaged melt rate estimates (L20), this low value for the 350-400m bin would pretty much suggest zero melt rates at the 350-400 depth given the much higher median melt rate from the previous 300-350m bin. Can you comment on either the reliability of this 350-400m estimate (there are zero error bars so presumably it is just one estimate) or whether this sudden drop may be meaningful in terms of a dramatic decrease in melt rate at a certain depth in Ilulissat Isfjord?

P5 re depth dependency and Fig. 3. In addition to the above, I think that suggesting that depth the dependency "is particularly pronounced for icebergs calved from the Upernavik glaciers (Fig. 3d) and Jakobshavn Isbræ (Fig. 3e)" is rather misleading. With the normalised data, it is perhaps most pronounced at Helheim and Zachariae. Based on Fig. 3a, one might argue that JI is the most atypical in the 200-400m depth bins.

P5, l22-24 – presumably extent of sea ice and melange are also relevant to the stratification and circulation over different timescales?

Fig. 4. I found this Figure extremely hard to interpret, in particular because it was very hard to see the gray-scale 'colour' of the year fill, especially when stars are used as symbols at JI and Koge Bugt. I would suggest changing the symbols so that they are all squares or circles (like Upernavik or Alison – check to see which looks better) and put the name of each glacier in the top left corner of each box (i.e. a) Kong Oscar through g) Koge Bugt).

In reality, there is so much complexity in the plots, I am not sure whether a seasonal or interannual pattern would be visible even if one were present. As such, I feel that the
line that "the lack of a coherent temporal signal across all study sites does not preclude the existence of temporal variations" is pretty much spot on. It would be good to see on a single graph, for your most frequently sampled glacier, a time series of melt-rate v time (on x-axis) for each 50m draft bin. I am sure that you have tried this but I just think it would be good to show, in a more visibly obvious way, that there is no clear seasonal or temporal pattern, something which I feel Fig. 4 fails currently to do.

Fig. 5. Again, I think the choice of stars as the symbol makes the gray scale fill very hard to see.

P7, l5-6. If you are not going to go in to the details of your velocity change calculation, you at least need to refer to the paper/equation/parameterisation that you use to make the claim that "the water velocity would need to increase from an average of approximately 0.05m/s to 0.3 m/s to produce the ∼0.12m/d to ∼0.46m/d increase in the area-averaged melt rate". It would also help to know what change in water temperature would also give the increased melt rate if you kept the velocity at 0.05 m/s?

P7, l7. Full stop after "hypothesis".

P7, l8. Better to say "..from Sermilik Fjord in south-east Greenland suggests that..."

---

## Author Response (AR1)

**Dear Editor.**

5

We have gone through the referee comments on "Greenland Iceberg Melt Variability from High-Resolution Satellite Observations" and have implemented a number of revisions. The manuscript has been noticeably improved and we thank the editor and both referees for their constructive comments.

The responses to the referees' comments are listed below. Referee comments are in black and our responses are in blue.

10 Thank you for considering the manuscript for publication, Ellyn Enderlin

**Reviewer #1 (Jason Amundson)**

Summary:

15 In this study the authors use digital elevation models derived from satellite imagery to investigate temporal and spatial observations in iceberg melt rates. The paper builds on previous work to provide estimates of iceberg melt rates across several fjords in Greenland, and demonstrates that iceberg melt rates depend on iceberg draft with the caveat that draft and melt rates are inferred from calculations of subaerial volume and assumed iceberg geometry. Meltwater from icebergs appears to be an important source of freshwater for fjords in Greenland, and

20 therefore this study has important implications for fjord circulation and submarine melting of glacier termini.

Major comments:

Most of my concerns with this paper are related to understanding the numerous sources of uncertainty that are inherent (and unavoidable) in the authors' calculations. It wasn't until I went back to re-read Enderlin and

25 Hamilton (2014) that I realized that these uncertainties had already been addressed in some detail previously. Therefore I think this paper would benefit from a 1-2 paragraph summary of the sources of error and their impact on the melt flux and melt rate calculations. Presumably this summary would be in Section 2. My sense is that the error in the melt flux calculations is small and that those calculations are therefore pretty robust. The depth-averaged melt rate calculations are more tenuous because they rely on an assumed iceberg geometry, which affects both the submerged surface area and the iceberg draft. 30

We have added a paragraph outlining the various uncertainty sources that can be quantified from the available data, with an additional note to call-out the fact that deviations in iceberg geometries from the assumed (cylindrical) shape cannot be quantified but should be taken into consideration when examining the data.

35 I would also feel more comfortable with the discussion of how submarine melt rates vary with depth if the paper more explicitly referred to the depth-averaged melt rates and drafts as proxies. For example, I believe that the draft is calculated with something like:  $h' = "draft proxy" = V_{sa}/A * \rho_i/(\rho_w-\rho_i),$

where h' is the draft proxy, V\_{sa} is the subaerial volume, A is the cross-sectional 40 area of the iceberg at the waterline, and \rho i and \rho ware the densities of ice and water. Including something along these lines would more precisely indicate what is actually being plotted in the various figures. Something similar could be written out to describe the proxy for the depth-averaged melt rate. Not sure if this correct, but:

1

 $m' = 1 / (h' * 2*pi*r) * dV_{sm}/dt$

where m' is the proxy for the depth-averaged melt, r is the average radius at the water line, and  $V_{sm}=V_{sa}^{+}$  (\rho\_i/(\rho\_w-\rho\_i) is the submerged volume. Depending on exactly how this calculation is made, you may be able to cancel out some terms.

- We agree with the reviewer that it is beneficial to include equations that clearly demonstrate how we use our surface observations to estimate the iceberg draft, submerged area, and melt rate. We have added the equations used to estimate draft and the area-averaged submarine melt rate. The melt rate equation also contains the equation for the submerged ice area. We did not adopt the specific term "proxy" in the text but made an effort to make it more clear that the draft, submerged area, and melt rate data are estimates that inherently have much larger uncertainties than the volume change estimates because the submerged shapes that we use to compute melt rates will most likely differ from the assumed cylindrical geometries.
  - My concern is that I'm just not sure how much faith to put in the melt rate vs. draft figures. There seems to be a pretty nice relationship between meltwater flux and submerged area (based on the assumed geometry). Is there a similarly nice relationship between meltwater flux and submerged volume? If not, maybe that can somehow be used to justify the choice of iceberg shape.
- The relationships between meltwater flux and submerged area and meltwater flux and iceberg volume are similar. We show the plots of meltwater flux versus submerged area because, as we state in the text, the slope of the best-fit lines can be used as an approximation of the melt rate. As such, we think that the meltwater flux versus submerged area plot is more helpful to show than plots of meltwater flux versus volume.
- 20

15

Also, in the discussion of the observed changes in melt rates at Jakobshavn (page 7), it would be nice to have the calculations of melt rates spelled out in more detail. How did you calculate the change in water velocity that would be needed to increase the melt rate? By how much would you have to change the water temperature to get a similar change in melt rate? Can you exclude our potential sources, such as an incursion of warm water? Did

- 25 the melange remain intact following the calving event? We agree with the reviewer that the discussion of melt rate change in Ilulissat Isfjord would benefit from the inclusion of iceberg thermodynamic equations. We have included the equations for turbulence- and buoyancy-driven submarine melt so that it is easier for the reviewer to assess the importance of temperature and velocity change as drivers of changes in melt rates. We point out that, in the absence of changes in relative velocity, the
- 30 temperature change required to drive the observed increase in deep-drafted icebergs melt rates is physically untenable given the range in water temperature observations from the fjord (see Gladish et al., 2015). These revisions should now make it more clear why we hypothesize that iceberg overturning essentially jump-started fjord circulation, which led to the four-fold increase in melt rates.

**35**

**Minor comments:**

p. 1, line 16: Consider pointing out in the abstract that you don't observe longitudinal variations in melt rates. That seems to be a pretty important finding. Added.

**40**

p. 1, line 30: Sublimation also contributes to ablation. Added.

p. 2, line 12: Why these seven glaciers?

On p 3. line 27 we have added that these sites were selected based on image availability. Specifically, we selected sites that spanned the majority of the ice sheet and had sufficient WorldView imagery to estimate iceberg melt rates over more than one observation period.

5 p.2, line 4: Consider citing Alon Stern's JGR paper: The effects of Antarctic iceberg calving-size distribution in a global climate model Added.

**p. 3, line 1: Why the switch in processing schemes?**

10 DEMs through 2014 were produced by Dr. Ian Howat using the SETSM algorithm that was in the final stages of development by his group at The Ohio State University at the time. Dr. Howat agreed to process the DEMs in exchange for information regarding SETSM DEM quality relative to DEMs produced using the NASA Ames Stereo Pipeline (ASP), which was in the process of being installed on the University of Maine's high-performance computing cluster. Following the installation of ASP, we compared several DEMs and found negligible differences in iceberg elevations, motivating us to switch to using ASP so that the DEMs could be produced entirely in-house on demand.

**p. 3, line 22: Are any of the icebergs tabular?**

For Zachariae Isstrom a large number of icebergs remained upright following their detachment from the glacier's floating ice tongue. Several of the deep-drafted (i.e., >250 m median keel depths) in the other fjords were tabular as well.

p. 5, line 22: Perhaps cite John Mortenson's paper: Heat sources for glacial melt in a sub-arctic fjord (godthabsfjord) in contact with the Greenland ice sheet

25 Added

**Reviewer #2 (Anonymous)**

- This paper investigates iceberg submarine melt variability in fjords for icebergs calved from seven large 30 tidewater glaciers around the Greenland coast between 2011-16. The paper uses a method developed and presented previously in a detailed paper (Enderlin and Hamilton 2014) and utilises Worldview Imagery to generate iceberg DEMs. The paper shows clearly how the estimated iceberg submarine melt-rates show distinct melt patterns that one would expect based both on hydrographic observations and variations in latitude and iceberg draft. In the main, the paper is very clearly written and the findings are well supported by the analyses 35 and data while the conclusion provides a very succinct and clear summary of the paper highlighting the key
- findings and the considerable potential of the method utilised.

There are a few areas where the paper is a little unclear and these are outlined below; in particular, some changes to the Figures would improve the clarity of the paper. On occasions, just a little more text is needed to aid the reader and any such additions will not detract from the paper as it is not overly long.

P1, 117 - I was a somewhat unclear what you meant in the abstract when stating that you do not resolve "coherent" temporal variations in melt rates. After reading the paper, this became clearer but I think it makes sense to state more clearly here in the abstract that you do resolve coherent 'seasonal or interannual patterns in your iceberg melt-rates'.

Changed.

45

P1, 127 – worth adding that the size distribution of the calved icebergs as well as the volume calved is crucial to the spatial distribution of iceberg freshwater fluxes.

5

P2, 117-18 – important to add the caveat here that this is true as long as the iceberg in question is floating. Revised for clarity.

P2, 121-23 – I think that you should also add other satellite platforms to your list of additional potential methods that could be used to derive elevation time-series.

Added.

Added.

P2, 131-32 – You state that "A comparison of the DEMs produced using the SETSM and ASP algorithms indicates that the accuracy of iceberg elevations is unaffected by the choice of the algorithm used to construct DEMs". You have presumably carried out some kind of analysis to demonstrate that this is the case; in which

- 15 DEMs". You have presumably carried out some kind of analysis to demonstrate that this is the case; in which case, it would be beneficial to report briefly (even in one line) what "unaffected" means by referring to one example of the results derived from analysis on one of your iceberg data-sets. We have changed the sentence so that it is more clear that a comparison of the datasets indicates that switching between the two algorithms does not introduce systematic biases into our results: "A comparison of the DEMs"
- 20 produced using the SETSM and ASP algorithms indicates that the accuracy of iceberg elevations derived from the algorithms are comparable, allowing us to switch from the use of SETSM DEMs for 2011-2014 images to ASP DEMs for 2015-2016 images without biasing our results." We do not show iceberg elevation maps or provide specific numbers for SETSM- and ASP-derived DEMs because a detailed cross-comparison is outside the scope of the manuscript. Noh and Howat (2015) discuss the quality of their algorithm and we refer the
- 25 reviewer to their manuscript and Shean et al. (2016) for details on the quality of ice DEMs produced using the ASP algorithm.

Fig. 1. Each location map (insets b - h) needs a scale (unless the scale is the same in all of them in which case a scale is still needed somewhere). Furthermore, it would help to show clearly where the calving fronts of the

- 30 glaciers are; this may be obvious in some figures either from visual clarity (g) or site familiarity (e) but for many, especially d, f and g, it's not really clear where the icebergs have come from. (I might add that it does become clear when I enhance the scale on the pdf to 400%, as the images are very high quality, they are just very small at the resolution of the current figure). And this information is needed to help make sense of the text on P3, 126-29.
- 35 All small panels have the same scale so a scalebar has been added to only panel b and a note about the scaling has been added to the legend. The terminus location in each panel has also been added.

Furthermore, in Figure 1 and in all subsequent figures, I think you should re-order the glacier legend box and symbols from b) to h) i.e. Kong Oscar, Alison, Upernavik etc ending with Koge Bugt rather than alphabetically as currently which is much more confusing for the reader.

Changed.

40

Fig. 2 and p4, l6. It would be very helpful to include the estimated submarine melt rates, derived from each linear polynomial, within the individual figure boxes (a - g).

4

45 Added

P4, L16. Ref. should be to Jackson and Straneo, not et al. Corrected.

P4, 118 – there are seven plotted symbols in Figure 1h) associated with Koge Bugt, not the six suggested in the 5 text. Why the discrepancy?

Thank you for pointing this out. There are seven icebergs and the text has been corrected.

P4, 121. The confidence for the Koge Bugt datasets, as currently explained, seems a little misplaced given the sample size. In particular, from Fig 1, it looks as though one of the icebergs sampled is a considerable distancefrom the others and perhaps in more open waters 'atypical' of the other fjord samples. As such, are you sure you

- are observing "typical melt conditions" and not getting spurious results due to anomalous sampling (particularly given the small sample size and thus significance of one anomalous data point to your overall results for KB).
   Although the one iceberg in the sample set is several kilometers down-fjord from the other icebergs, this is not one of the icebergs with the exceptionally high melt rates. Also, it is important to note that all the small panels in
   Figure 1 have the same scaling, so the iceberg located farthest from the terminus is actually not exceptionally far
- 15 Figure 1 have the same scaling, so the iceberg located farthest from the terminus is actually not exceptionally far away from the glacier relative to observations from other fjords. Furthermore, we are confident that our interpretation is not skewed by one anomalous data point that is not representative of melt conditions near the terminus because there is actually a deep-drafted iceberg with a melt rate of ~0.75 m/d during each observation period.
- 20

P4, 131 – will detailed in-situ data become more widely available as part of the OMG programme?

Yes, the number of in situ hydrographic observations available around Greenland will drastically increase as a result of NASA's OMG program. Future research will explore whether there are any coincident in situ hydrographic observations and remotely-sensed iceberg melt rates.

P5, 18 - The results for individual glaciers (b-h) would be of much more use if the y scale was reduced from 0 - 0.4 m/d as opposed to 1m (with the exception of Koge Bugt) so that the (valuable) details in the variable meltrates could be seen more clearly.

- 30 We agree that the variability with depth is hard to discern in panels b-h, which is why we have also included the normalized data in panel a. We have modified the y-scaling so that it now goes from 0-0.9 m/d but choose to keep the same scaling for panels b-h so that the melt magnitudes can be directly compared between panels.
- P5, L12-13. The broad description relating to melt rate with iceberg draft is not really correct when integrating the Upernavik and Jakobshavn Isbrae results. The melt rate actually continues to decrease in draft bin 200-250m at JI, not "increase" again as suggested in L13. Hence the dip is broader than the 150-200m dip that you suggest. I think you just need to be a bit broader in your depth categorisation for the "approximate depth of the interface" for the cold-warm boundary (and I presume that it does vary between fjord systems). Furthermore, the melt rate appears to dip again at JI in the 350-400 m bin (actually dropping back to shallow 'cold' water values). Given
- 40 the integrative nature of your area averaged melt rate estimates (L20), this low value for the 350-400m bin would pretty much suggest zero melt rates at the 350-400 depth given the much higher median melt rate from the previous 300-350m bin. Can you comment on either the reliability of this 350-400m estimate (there are zero error bars so presumably it is just one estimate) or whether this sudden drop may be meaningful in terms of a dramatic decrease in melt rate at a certain depth in Ilulissat Isfjord?
- 45 We have modified this section slightly to reflect the observation that the dip in melt rates extends to a deeper water depth in Ilulissat Isfjord than in the Upernavik region. We also point out that the melt rate estimate for

350-400 m-depth is based on one observation from March 2011. As discussed in section 3.3, the March 2011 melt rates in Ilulissat are exceptionally low and this low melt rate estimate should not be considered as an indication that melting actually ceases below 350 m-depth.

- 5 P5 re depth dependency and Fig. 3. In addition to the above, I think that suggesting that depth the dependency "is particularly pronounced for icebergs calved from the Upernavik glaciers (Fig. 3d) and Jakobshavn Isbræ (Fig. 3e)" is rather misleading. With the normalised data, it is perhaps most pronounced at Helheim and Zachariae. Based on Fig. 3a, one might argue that JI is the most atypical in the 200-400m depth bins.
- We agree that the normalized data show marked increases in melt rates for Helheim and Zachariae below ~200 10 m-depth but the actual magnitude of the increase with depth is much smaller for Zachariae (0.01 to 0.1 m/d) than for Upernavik (0.03 to 0.26 m/d) and Jakobshavn (0.11 to 0.41m/d). Although the magnitude of the melt rate change with depth is comparable to Helheim (0.16 to 0.32 m/d), the depth dependency of the melt rate is less pronounced for individual observation periods (see Fig 4) than observed for Upernavik and Jakobshavn. Hence our focus on Upernavik and Jakobshavn in the paper.
- 15

20

**P5, 122-24 – presumably extent of sea ice and melange are also relevant to the stratification and circulation over different timescales?**

Sea ice and ice mélange extent likely influence the wind stress exerted on the surface water layer, as well as the temperature and salinity of surface and possibly near-surface waters. As such, we have added that they likely influence fjord circulation, with references to papers describing mélange meltwater fluxes in Helheim and Jakobshavn's fjords (Enderlin et al., 2016) and the influence of sea ice extent on melting of Petermann Glacier

- in northern Greenland (Shroyer et al., 2017), as supporting examples. Fig. 4. I found this Figure extremely hard to interpret, in particular because it was very hard to see the gray-scale 'colour' of the year fill, especially when stars are used as symbols at JI and Koge Bugt. I would suggest
- 25 'colour' of the year fill, especially when stars are used as symbols at JI and Koge Bugt. I would suggest changing the symbols so that they are all squares or circles (like Upernavik or Alison check to see which looks better) and put the name of each glacier in the top left corner of each box (i.e. a) Kong Oscar through g) Koge Bugt).
- In reality, there is so much complexity in the plots, I am not sure whether a seasonal or
- 30 interannual pattern would be visible even if one were present. As such, I feel that the line that "the lack of a coherent temporal signal across all study sites does not preclude the existence of temporal variations" is pretty much spot on. It would be good to see on a single graph, for your most frequently sampled glacier, a time series of melt-rate v time (on x-axis) for each 50m draft bin. I am sure that you have tried this but I just think it would be good to show, in a more visibly obvious way, that there is no clear seasonal or temporal pattern, something 35 which I feel Fig. 4 fails currently to do.
- We agree that the fill color of the star symbols was difficult to see and we have enlarged the symbols so that the color difference is more obvious. We have kept the different symbols, however, because they are intended to help colorblind readers distinguish the different fjord locations in the normalized melt rate plot (Fig 3a). We have also plotted time series of the binned melt rates, as shown below, to support our interpretation that there are
- 40 no consistent temporal variations in melt rates across all study sites. In the figure, the panels are arranged in the same order as Figures 1-4 (panel locations in the approximate locations of the study sites). The marker edge and line colors distinguish the observation depths, with the colors gradually transitioning from black for the near-surface observations to orange at 350-400 m-depth. In this figure, it is clear that the differences in the periods of observation obscure any temporal patterns across all sites. We have chosen not to add this figure to the paper
- 45 because we feel it is unnecessary to include two plots addressing the same data, particularly because it

demonstrates a null result. However, if the editor feels strongly that this plot should be included, we can finish formatting this figure and include it in the text.

5 Fig. 5. Again, I think the choice of stars as the symbol makes the gray scale fill very hard to see. We have enlarged the symbols so they are easier to see. Since the focus of this plot is point out the rapid change in melt rates from March to April in 2011, which we call-out in boxes, we think that enlarging the symbols is a sufficient modification to address the reviewer's comment.

P7, 15-6. If you are not going to go in to the details of your velocity change calculation, you at least need to refer to the paper/equation/parameterisation that you use to make the claim that "the water velocity would need to increase from an average of approximately

0.05m/s to 0.3 m/s to produce the \_0.12m/d to \_0.46m/d increase in the area-averaged melt rate". It would also 5 help to know what change in water temperature would also give the increased melt rate if you kept the velocity at 0.05 m/s?

We have added iceberg melt equations here and elaborated on why we hypothesize that changes in velocity primarily drove the observed increase in the melt rate. Namely, temperature variations within the range of observed temperatures in Ilulissat Isfjord are not sufficient to drive such a large increase in melting. This should
hopefully be more clear with the text revisions and inclusion of the melt equations.

8

P7, 17. Full stop after "hypothesis". Changed.

15 P7, 18. Better to say "..from Sermilik Fjord in south-east Greenland suggests that. . ." Changed.

**Greenland Iceberg Melt Variability from High-Resolution Satellite Observations**

Ellyn M. Enderlin1,2, Caroline J. Carrigan2, William H. Kochtitzky1,2, Alexandra Cuadros3, Twila Moon4, Gordon S. Hamilton1,2,\*

1Climate Change Institute, University of Maine, Orono, ME, USA 04469
 2School of Earth and Climate Sciences, University of Maine, Orono, ME, USA 04469
 3School of Marine Sciences, University of Maine, Orono, ME, USA 04469
 4National Snow and Ice Data Center, University of Colorado, Boulder, CO, USA 80303
 \*Deceased

Correspondence to: Ellyn M. Enderlin (ellyn enderlin@gmail.com)

Abstract. Iceberg discharge from the Greenland Ice Sheet accounts for up to half of the freshwater flux to surrounding fjords and ocean basins, yet the spatial distribution of iceberg meltwater fluxes is poorly understood. One of the primary limitations for mapping iceberg meltwater fluxes, and changes over time, is the dearth of iceberg submarine melt rate estimates. Here
 we use a remote sensing approach to estimate submarine melt rates during 2011-2016 for 637 icebergs discharged from seven marine-terminating glaciers fringing the Greenland Ice Sheet. We find that spatial variations in iceberg melt rates generally follow expected patterns based on hydrographic observations, including a decrease in melt rate with latitude and an increase in melt rate with iceberg draft. However, we find no longitudinal variations in melt rates within individual fjords.

We do not resolve coherent seasonal to interannual patterns in melt rates across all study sites, though we attribute a fourfold melt rate increase from March to April 2011 near Jakobshavn Isbræt to fjord circulation changes induced by the seasonal onset of iceberg calving. Overall, our results suggest that remotely-sensed iceberg melt rates can be used to characterize spatial and temporal variations in oceanic forcing near often inaccessible marine-terminating glaciers.

**1** Introduction**

10

- The Greenland Ice Sheet discharges ~550 Gt of icebergs per year (Enderlin et al., 2014). This accounts for approximately a third to a half of the total freshwater flux from Greenland to the surrounding fjords and ocean basins (Bamber et al., 2012; Enderlin et al., 2014; van den Broeke et al., 2016). Unlike surface meltwater runoff fluxes from the ice sheet and tundra, which primarily enter the ocean system from point sources (subglacial discharge channels and terrestrial rivers, respectively), icebergs act as distributed freshwater sources. The spatial distribution of iceberg freshwater fluxes is dependent on a number of factors, including the volume and size distribution of ice calved from each glacier, which varies
- 30 substantially over a range of spatial scales (Enderlin et al., 2014), and the solid-to-liquid conversion rate of an iceberg's freshwater reserves. Although surface sublimation and melting, wave erosion, and submarine melting all contribute to iceberg ablation, the solid-to-liquid conversion rate should primarily be dictated by submarine melting because of the strong dependence of total ablation on the surface area over which each process acts (e.g., Enderlin et al. (2016), Moon et al. (2017)).

1

Ellvn Enderlin 11/18/2017 5:13 AM Deleted: large Ellyn Enderlin 11/18/2017 5.14 AM Deleted: general Ellyn Ende lin 11/18/2017 6: Deleted: temporal variations Ellvn Enderlin 11/18/2017 5:16 AM Deleted: iceberg Ellyn Enderlin 11/18/2017 5:16 AM Deleted: in iceberg melt rates Ellyn Enderlin 11/18/2017 5:16 AM Deleted: 's terminus Ellyn Enderlin 11/18/2017 5:11 AM Deleted: rapid Ellvn Enderlin 11/18/2017 5:17 AM Deleted: , 
[revised manuscript text omitted]

8

| Ellyn Enderlin 11/17/2017 7:17 PM                |
|--------------------------------------------------|
| Deleted: Assuming a relatively cool and          |
| Ellyn Enderlin 11/18/2017 5:04 AM                |
| Deleted: constant                                |
| Ellyn Enderlin 11/18/2017 5:03 AM                |
| Deleted: subsurface                              |
| Ellyn Enderlin 11/18/2017 5:03 AM                |
| Deleted: in March-April 2011                     |
| Ellyn Enderlin 11/18/2017 5:04 AM                |
| Deleted: water                                   |
| Ellyn Enderlin 11/18/2017 5:05 AM                |
| Deleted: produce the                             |
| Ellyn Enderlin 11/18/2017 5:05 AM                |
| Deleted: increase in the area-averaged melt rate |
| Ellyn Enderlin 11/19/2017 11:57 AM               |
| Deleted: , h                                     |
| Ellyn Enderlin 11/19/2017 11:57 AM               |
| Deleted: the                                     |

(4)

[revised manuscript text omitted]

---

## Referee Report (RR1)

I have now read the revised manuscript, and would like to thank the authors for their careful responses to my review. I have a few remaining and related concerns/suggestions related to melt vs. draft figures.

1. I think that one needs to be very careful when interpreting the variations in area-averaged melt rate with draft presented in Figures 4-6. The crux of this problem is that the submerged iceberg areas and drafts are unknown and difficult to determine. I think the approach of calculating a mean draft makes sense, and actually you can show that Equation 1 is valid for any iceberg in which the submerged volume of ice lies below the horizontal cross-sectional area of the iceberg at sea level (i.e., not just for the cylindrical geometry that the authors assume). The bigger issue is that calculating an area-averaged melt rate requires making an assumption about the iceberg geometry. Enderlin and Hamilton (2014) addressed this issue by considering both cylindrical and cone-shaped icebergs, and found that the assumed geometry affects the area-averaged melt rate by about 10%. Thus it would seem that the assumed geometry doesn't affect the results too much, although I'm not entirely sure if that's true. What I'm having trouble with is that, if you assume that the icebergs are always cylindrical, then you are in essence assuming that the submarine melt rate doesn't vary with depth — and that makes it dificult to see why the area-averaged melt rate should depend on draft... I wonder if it would make more sense to plot the fractional rate of volume change (1/V * dV/dt) vs. draft as that would also be useful for assessing how freshwater fluxes vary with depth and wouldn't require any assumptions about iceberg geometry.

2. Related to item (1), some of the individual data points in Figures 4-6 are based on very small statistics, which makes it difficult to assess the significance of the trends. It may be nice to have a table or figure that somehow illustrates the size distribution of the icebergs that were analyzed, or somehow modify the figures to indicate how many samples are included in each data "bin". A sentence or two in the text may also suffice.

3. The dips in melt rate observed at Upernavik and Jakobshavn (Fig. 3d-e) are within the error bars of the adjacent points, and so the dips in melt rate may not be significant. That should be made clear in the text.

4. Some of the sharp changes in area-averaged melt rates seem counterintuitive. For example, the rapid increase in melt rate at Koge Bugt (Fig. 3h) between 210 and 290 m depth would seem to indicate that the average melt rate over that depth range is something like 2.5 m/d, or more than an order of magnitude larger than the average melt rate over the upper 210 m. The same could also be said for the increase in melt rate between 140 m and 170 m depth at Jakobshavn Isbrae in April 2011 (Fig. 5). Is it plausible to have such sharp changes in melt rates over these distances?

5. I feel much more comfortable with the general trends presented in Figure 3, which seem to be more statistically significant. Figure 3a is especially nice.

---

## Author Response (AR2)

Dear Editor,

We have gone through the comments made by you and the reviewer on "Greenland Iceberg Melt Variability from High-Resolution Satellite Observations" and have implemented appropriate revisions. We hope that the manuscript is now suitable for publication.

The responses to the comments are listed below. Comments are in black and our responses are in blue.

Thank you for considering the manuscript for publication,
Ellyn Enderlin

**Second review comments:**
1. I think that one needs to be very careful when interpreting the variations in area-averaged melt rate with draft presented in Figures 4-6. The crux of this problem is that the submerged iceberg areas and drafts are unknown and difficult to determine. I think the approach of calculating a mean draft makes sense, and actually you can show that Equation 1 is valid for any iceberg in which the submerged volume of ice lies below the horizontal cross-sectional area of the iceberg at sea level (i.e., not just for the cylindrical geometry that the authors assume). The bigger issue is that calculating an area-averaged melt rate requires making an assumption about the iceberg geometry. Enderlin and Hamilton (2014) addressed this issue by considering both cylindrical and cone-shaped icebergs, and found that the assumed geometry affects the area-averaged melt rate by about 10%. Thus it would seem that the assumed geometry doesn't affect the results too much, although I'm not entirely sure if that's true. What I'm having trouble with is that, if you assume that the icebergs are always cylindrical, then you are in essence assuming that the submarine melt rate doesn't vary with depth and that makes it difficult to see why the area-averaged melt rate should depend on draft...
I wonder if it would make more sense to plot the fractional rate of volume change (1/V *dV/dt) vs. draft as that would also be useful for assessing how freshwater fluxes vary with depth and wouldn't require any assumptions about iceberg geometry.
We agree that the uncertainty in the submerged geometry is the largest source of uncertainty in the area-averaged melt rate estimates. It is true that variations in iceberg melting with depth will lead to deviations in iceberg shape from the assumed, cylindrical, geometry. However, the variations in melt rate with depth should not lead to marked changes in submerged area, and thus average melt rate, over the relatively short time periods considered here (average of ~20 days). For example, if the melt rate increases linearly from a minimum of 0.1 m/d near the surface to a maximum of ~0.6 m/d at the base of the largest icebergs in our study (~500 m-width, ~350 m-depth), so that the area-averaged melt rate is ~0.4 m/d (as near the termini of Jakobshavn Isbræ and Helheim Glacier), then submarine melting will decrease the submerged iceberg area by ~6% over the typical 20-day observation period. The average uncertainty in our iceberg submerged area estimates, which we estimate from temporal variations in the iceberg surface elevation and plan-view surface area, is ~5%. Thus, we feel that our uncertainty estimates reasonably capture the uncertainty in iceberg geometry and area-averaged melt rate that results from the evolution of iceberg geometry below the waterline.

Recently-published iceberg modeling work performed by co-author Moon ("Subsurface iceberg melt key to Greenland fjord freshwater budget", Nature Geoscience, doi:10.1038/s41561-017-0018-z) provides additional support for the use of a constant iceberg shape over the relatively short time periods considered herein: over the 1-month timestep used in her model, the iceberg geometries changed so little that she could have used a constant geometry with no significant impact on her freshwater flux estimates.

We have, however, constructed a figure of normalized meltwater flux (i.e., volume change over time divided by the average volume) as suggested by the reviewer. Vertical uncertainty bars are omitted in the resultant figure below to highlight the variations (or lack thereof) in normalized meltwater flux with iceberg draft. The data suggest that the normalized meltwater flux varies little with draft, which we interpret as additional support for our observation of increased melt rates with depth. If the melt rates were uniform with depth, then we would expect to see pronounced negative slopes in the figure below because of the decrease in the submerged area-to-volume ratio with increasing iceberg size. Therefore, the lack of a large and consistent decrease in normalized meltwater flux with depth is suggestive of an increase in melt rate with depth.

We have added several sentences to the text to clarify the aforementioned points. Specifically, we call attention to the relatively small impact that depth-dependent melt rates have on the submerged iceberg geometry over sub-monthly time scales.

[Figure]

2. Related to item (1), some of the individual data points in Figures 4-6 are based on very small statistics, which makes it difficult to assess the significance of the trends. It may be nice to have a table or figure that somehow illustrates the size distribution of the icebergs that were analyzed, or somehow modify the figures to indicate how many samples are included in each data "bin". A sentence or two in the text may also suffice.
The number of observations per date, per bin are now listed in Table 1.

3. The dips in melt rate observed at Upernavik and Jakobshavn (Fig. 3d-e) are within the error bars of the adjacent points, and so the dips in melt rate may not be significant. That should be made clear in the text.
The text has been revised to make this clearer.

4. Some of the sharp changes in area-averaged melt rates seem counterintuitive. For example, the rapid increase in melt rate at Koge Bugt (Fig. 3h) between 210 and 290 m depth would seem to indicate that the average melt rate over that depth range is something like 2.5 m/d, or more than an order of magnitude larger than the average melt rate over the upper 210 m. The same could also be said for the increase in melt rate between 140 m and 170 m depth at Jakobshavn Isbrae in April 2011 (Fig. 5). Is it plausible to have such sharp changes in melt rates over these distances?
Although the changes in melt rate with depth seem quite large for some time periods, the changes are actually more gradual than suggested by the reviewer. Since submarine melting occurs over both the lateral margins and the base of the iceberg and the area of the iceberg base is typically comparable to that of the lateral margins (Enderlin et al. (2016) show the W:H ratio for icebergs calved from Jakobshavn and Helheim is ~1.5), abrupt increase in melt rate with depth do not necessarily require order-of-magnitude changes in the melt rate. For example, if the melt rate is ~0.2 m/d down to ~125 m-depth (as near Jakobshavn's terminus in April 2011), then a melt rate of ~0.8 m/d below the Polar-Atlantic water interface will lead to a depth-averaged melt rate of ~0.5 m/d. Comparable variations in melt rate with depth have been empirically estimated near Helheim's terminus (see Moon et al. (2017) Figure 2), providing independent support that such drastic changes in melt rate are plausible.

5. I feel much more comfortable with the general trends presented in Figure 3, which seem to be more statistically significant. Figure 3a is especially nice.
Thank you.

*Additional comments:*
Figure 2 caption: add a statement about m, i.e. submarine melt rates derived from each linear polynominal is given in the box.
Added.

Figure 5: Do you want to say in the caption "Highly variable melt rates at greater depths observed in March and April 2011 are highlighted with the boxes."? "Deep melt rates" is confusing and there is no circle. Also, consider showing the rate variations using arrows to clarify the temporal change, instead of highlighting boxes. Similar to reviewer #2, I also feel that it is hard to read Figure 5, even in the revised version. Make this figure tall enough in the final manuscript (one column width is enough).
The figure has been made taller and narrower to fit in one column and to emphasize the difference in melt rates over time. We have changed the rectangular bounding boxes so that they are now shaded boxes that encompass the range of melt rates for deep-drafted icebergs for each observation period. The average melt rate is now marked on the plot and the change in the average melt rate is marked by a labeled double-sided arrow. The legend has also been revised accordingly.

Figure 3: include the value of the observed melt rate in the caption.

Included.

Table 1: change the order of rows so that these datasets are listed in the order same as Figure legends (i.e. Kong Oscar first, Koge Bugt last).

Changed.

Table 1: add a summary sentence at the beginning, e.g. Observation overview and derived meltwater flux.

Added

[revised manuscript text omitted]